# MAD-UNet: A Multi-Region UAV Remote Sensing Network for Rural Building Extraction

**DOI:** 10.3390/s24082393

**Published:** 2024-04-09

**Authors:** Hang Xue, Ke Liu, Yumeng Wang, Yuxin Chen, Caiyi Huang, Pengfei Wang, Lin Li

**Affiliations:** 1School of Remote Sensing and Information Engineering, North China Institute of Aerospace Engineering, Langfang 065000, China; xuehang@stumail.nciae.edu.cn (H.X.); wym@stumail.nciae.edu.cn (Y.W.); cyx1114@stumail.nciae.edu.cn (Y.C.); huangcy@stumail.nciae.edu.cn (C.H.); wpf544943@stumail.nciae.edu.cn (P.W.); lilin@stumail.nciae.edu.cn (L.L.); 2Hebei Collaborative Innovation Center of Space Remote Sensing Information Processing and Application, Langfang 065000, China

**Keywords:** deep learning, U-Net, unmanned aerial vehicle remote sensing images, rural buildings

## Abstract

For the development of an idyllic rural landscape, an accurate survey of rural buildings is essential. The extraction of rural structures from unmanned aerial vehicle (UAV) remote sensing imagery is prone to errors such as misclassifications, omissions, and subpar edge detailing. This study introduces a multi-scale fusion and detail enhancement network for rural building extraction, termed the Multi-Attention-Detail U-shaped Network (MAD-UNet). Initially, an atrous convolutional pyramid pooling module is integrated between the encoder and decoder to enhance the main network’s ability to identify buildings of varying sizes, thereby reducing omissions. Additionally, a Multi-scale Feature Fusion Module (MFFM) is constructed within the decoder, utilizing superficial detail features to refine the layered detail information, which improves the extraction of small-sized structures and their edges. A coordination attention mechanism and deep supervision modules are simultaneously incorporated to minimize misclassifications. MAD-UNet has been tested on a private UAV building dataset and the publicly available Wuhan University (WHU) Building Dataset and benchmarked against models such as U-Net, PSPNet, DeepLabV3+, HRNet, ISANet, and AGSCNet, achieving Intersection over Union (IoU) scores of 77.43% and 91.02%, respectively. The results demonstrate its effectiveness in extracting rural buildings from UAV remote sensing images across different regions.

## 1. Introduction

The construction of beautiful rural areas is a significant aspect of China’s rural revitalization strategy. Accurate surveying and investigation of rural buildings facilitate more effective rural planning and the development of livable and productive, beautiful rural areas [1]. The primary methods for extracting buildings from high-spatial-resolution remote sensing images are categorized into classical computer vision methods and machine learning methods. Classical computer vision methods, such as edge detection [2] and region-based segmentation algorithms [3], primarily extract buildings by analyzing intuitive features like shape, spectrum, and texture in remote sensing images. Meanwhile, machine learning methods, including random forests [4] and support vector machines (SVMs) [5], automatically identify buildings by learning complex features from image data. While these methods can effectively identify and learn from large datasets, their success is highly reliant on domain expertise in remote sensing and extensive practical experience in feature selection. With high-spatial-resolution remote sensing images becoming ever more complex and detailed, both classical computer vision and machine learning methods encounter significant challenges in enhancing extraction accuracy, automation, and processing speed [6].

In recent years, deep learning algorithms have gained prominence in applications such as image classification, object detection, and semantic segmentation [7]. Their powerful feature extraction and recognition capabilities have led to revolutionary progress in the automatic extraction of ground information from remote sensing images. In 2015, Long et al. [8] introduced a seminal work in the field of semantic segmentation with the Fully Convolutional Network (FCN), which replaced the fully connected layers in traditional Convolutional Neural Networks (CNNs) with convolutional layers, achieving pixel-level classification of images. The field has since seen the emergence of a series of classic networks, such as U-Net [9], PSPNet [10], and the DeepLab series [11,12,13]. These networks can autonomously learn complex spectral and texture features in remote sensing images and automatically extract features of buildings, roads, and other targets [14]. They have improved segmentation accuracy through various means, including the skip-connection structure in U-Net that retains details, the pyramid pooling structure in PSPNet that aggregates contextual information at multiple scales, and the Atrous Spatial Pyramid Pooling (ASPP) module in DeepLab V3+, which uses atrous convolution to expand the receptive field [15]. Numerous scholars have built upon the design concepts of these traditional networks and proposed new network architecture modules to enhance semantic segmentation performance for specific tasks. Gong et al. [16] designed a context-aware module that captures long-range dependencies between building locations using a self-attention mechanism while extracting multi-scale building features, thus enhancing the model’s capacity to represent building features. Yu et al. [17] introduced an attention gate module added to the skip-connection structure of U-Net, which suppresses irrelevant features and noise in the input image, emphasizing the primary features of buildings. Further developing U-Net, Yu et al. [18] created the CC-Net module to connect the encoder and decoder, capturing essential building features, adding multi-scale semantic information to the network, and using multiple attention gates to model global dependencies between buildings. Wang et al. [19] presented a global feature information perception module that addresses the semantic gap between low-level and high-level features and utilizes a spatial attention mechanism, improving the model’s interpretation of spatial feature information. Compared to the CNN-based methods, Transformer-based methods are more adept at handling long-range dependencies in images. Gibril et al. utilized the self-attention mechanism in Transformers, which allows the model to consider information from the entire image when processing each image patch, effectively capturing global dependencies [20]. Chen et al. proposed a Transformer-based dual-branch network called ADF-Net, which not only enhances the recognition of building features and the expression of spatial information but also captures global dependencies between buildings effectively [21]. Although Transformer-based methods have shown strong performance in image segmentation tasks, they generally require more computational resources and larger training datasets to achieve optimal performance. Therefore, when selecting a suitable method for a specific remote sensing image processing task, it is important to consider the model’s complexity, training cost, and execution efficiency comprehensively. Future research could explore combining Transformers and CNNs and finding ways to reduce their computational demands while maintaining model performance. This will be crucial for achieving efficient and accurate surveying of rural buildings.

Although numerous researchers have investigated the semantic segmentation of buildings in high-spatial-resolution remote sensing images and have achieved favorable results in urban building extraction, there is limited research on the extraction of rural buildings in different provinces of China, which feature various construction styles [22]. Compared to urban areas, rural buildings in different regions exhibit diverse backgrounds, and building structures and layouts can vary significantly due to differences in geographical landscapes and cultural practices. This variation presents challenges in automatic building extraction [23,24] and may lead to issues such as omissions, incorrect extractions, and unclear boundaries between buildings [25]. To address these issues and challenges, this study proposes an improved network based on U-Net, the Multi-attention-Detail U-shaped Network (MAD-UNet), which integrates multi-scale contextual information with detail feature enhancement through the design of Multi-scale Fusion Modules and Detail Feature Extraction Modules (DFEM). This integration optimizes the extraction of building edges and addresses the issue of building boundary adhesion in the extraction of rural buildings across different regions. Additionally, a pyramid pooling structure with atrous convolution is introduced to enhance the feature extraction ability of the backbone network. A Cascade Multi-Scale Coordinate Attention (CMSCA) module is designed to enhance attention to multi-scale semantics and spatial information, thereby reducing the occurrence of missed and incorrect segmentations during the extraction of rural buildings in various regions. Furthermore, deep supervision is proposed, supervising different levels of features within the decoder to improve the model’s robustness. This approach makes the model more suitable for tasks such as building recognition and segmentation in high-spatial-resolution drone remote sensing images with a spatial resolution of 0.5 m. The main contributions of this research can be summarized as follows:(1)We constructed a building image dataset for typical rural areas in China, encompassing complex extraction scenarios across multiple provinces and various types of buildings with different layouts;(2)A novel skip connection structure, CMSCA, has been proposed. This structure extracts building features from different layers at multiple scales and enhances the global relevance of the model. It more effectively captures the dependencies between different positions within the input feature map, emphasizing the model’s focus on crucial information;(3)A new method for detail feature enhancement has been proposed, involving the reutilization of shallow detail features from the U-Net encoder and the fusion of them with edge information extracted using the Sobel operator in the decoder. This enhances the model’s ability to acquire knowledge about edges and detailed information;(4)MAD-UNet has been tested on a proprietary unmanned aerial vehicle dataset and the publicly available Wuhan University (WHU) Building Dataset, demonstrating the rationality and effectiveness of the improved algorithm. The study provides a scientific reference for extracting rural buildings from high-spatial-resolution remote sensing images.

The main sections of this article are arranged as follows: Section 2 provides a detailed introduction to the overall architecture of the MAD-UNet network, the design of the Multi-scale Feature Fusion Module (MFFM), the design of the CMSCA module, and the incorporation of the ASPP module and deep supervision mechanism. Section 3 describes the collection and construction of experimental data and the selection of evaluation metrics. Section 4 presents the experimental parameter settings, displays the experimental results, and provides a comparative analysis. Section 5 summarizes the advantages of the network and prospects for future work.

## 2. Materials and Methods

### 2.1. MAD-UNet Network Architecture

The MAD-UNet, building on the U-Net architecture, comprises a multi-scale fusion module, a detail feature enhancement module, a CMSCA module, and a deep supervision module. The overall structure of the network is depicted in Figure 1. The multi-scale fusion module is designed to integrate feature information from various scales and levels, enabling a more refined abstraction of building features. The detail feature enhancement module uses the Sobel operator and a gated unit mechanism to extract and refine building edge information, thereby improving the network’s ability to capture detailed building information. Furthermore, the CMSCA module replaces U-Net’s original skip connection structure, aiming to extract multi-scale information from feature maps and establish long-range dependencies between feature map pixels. This aids the model in better understanding pixel relationships.

MAD-UNet comprises two parts: an encoder and a decoder. The encoder uses ResNet50 as the backbone feature extraction network, with the initial 7 × 7 convolution, max pooling, and the final global average pooling and fully connected layers removed. The first encoding layer is typically responsible for capturing certain local detail features of buildings, such as edges, textures, and colors. To enhance the local detail feature information of buildings and improve the edge extraction effect, MAD-UNet introduces a detail feature enhancement module. This aims to reuse local feature details from shallow feature layers and fuse them with certain deep features. Due to repeated downsampling operations, the feature map’s receptive field increases, but many feature details are lost, and it is difficult to recover global semantic information using upsampling alone. Therefore, in the skip connections, MAD-UNet employs a CMSCA module to enhance the model’s semantic expressive capacity for feature extraction, optimize the semantic differences between shallow and deep feature maps, suppress the introduction of noise information, and minimize misclassification during multi-scale building extraction. Lastly, to address the insufficient modeling of building semantic information and multi-scale information in U-Net, the common ASPP structure is added between the encoder and the decoder. This addition augments the extraction of multi-scale feature information of buildings in the backbone feature extraction network. Specific parameters of MAD-UNet are shown in Table 1.

### 2.2. ASPP Structure 

In remote sensing imagery, the variance in building sizes can lead to the omission of small buildings during semantic segmentation. To improve the model’s ability to recognize buildings of varying scales, an ASPP module is incorporated between the encoder and decoder. Atrous convolution is employed to expand the convolution kernel’s effective field of view [26], while parallel multi-branch structures enhance contextual semantic information, thus improving segmentation performance [27]. However, overly large atrous rates may lead to the loss of local feature information [28]. Consequently, smaller atrous rates (1, 3, 5, 7) were selected for the atrous convolutions to better capture building features across different scales and enhance the network’s contextual awareness. Drawing inspiration from the residual structure of ResNet [29], a residual structure has been incorporated into the ASPP, as illustrated in Figure 2. This approach allows the network to learn new features while preserving low-level information, preventing the loss of bottom-layer features and reducing the risk of exploding or vanishing gradients during the training of deep networks.

### 2.3. MFFM Module

In the U-Net encoder, multiple pooling operations are used to expand the receptive field, which enhances the model’s capability to represent global features and semantics. However, these operations also reduce the spatial resolution of the image. Consequently, effectively recovering the spatial information lost during the downsampling process is difficult when relying solely on upsampling in the decoder [30,31]. To improve U-Net’s ability to reconstruct the local structure and fine details of buildings during the upsampling process, an MFFM and a Detail Feature Extraction Module (DFEM) have been designed. With the exception of MFFM1, which retains only the skip-connection part, the overall flow of a single MFFM is shown in Figure 3. Here, $E_{i},$ where i∈(2,3,4), represents the features from the 2nd, 3rd, and 4th layers of the encoder. $D_{i},$ where i∈(2,3,4), corresponds to the decoder feature layers associated with $E_{i}$, and $F_{i},$ where i∈(2,3,4), denotes the detail features extracted from $E_{i}$. $E_{i}$ and $D_{i}$ are linked by skip connections, then multiplied and added to $F_{i}$. The optimized features then proceed to the next MFFM module.

To enhance the model’s learning capability for building boundaries and local texture features, a Detail Enhancement Module was designed to amplify the detail features from the shallow layers in the encoder and to integrate these optimized features with the features from various levels within the decoder. The workflow of the Detail Enhancement Module is depicted in Figure 4, comprising two branches. The shallow features within the encoder capture the most basic details, such as edges, textures, and colors. In the first branch, the first-layer feature E_1_ of the encoder is processed through three sets of convolutions of different sizes to extract multi-scale detail features. The feature map E_1_ first goes through a 5 × 5 convolution to obtain feature F_1_, which is then added to E_1_ and further processed through a 3 × 3 convolution to obtain feature F_2_. Subsequently, F_2_ is added to F_1_, and the result undergoes a 1 × 1 convolution in the third branch, resulting in feature F_3_. Features F_1_, F_2_, and F_3_ are concatenated to obtain the multi-scale detail features F_4_. Finally, F_4_ is processed through convolutional layers consisting of a set of three 3 × 3 convolutions, two sets of double 3 × 3 convolutions, and three sets of triple 3 × 3 convolutions, to yield features ${\;M}_{i}$, where i∈(1,2,3). The specific calculation process is as follows: (1)$$F_{1}=Conv_{5\;\times 5}\left(E_{1}\right)$$
(2)$$F_{2}=Conv_{3\times 3}\left(E_{1}+F_{1}\right)$$
(3)$$F_{3}=Conv_{1\times 1}\left(F+F_{2}\right)$$
(4)$$F_{4}=Concat\left(F_{1}+F_{2}+F_{3}\right)$$
where *Conv* represents the convolution operation and *Concat* represents the concatenation operation.

Due to the Sobel operator also extracting edge information of ground objects other than buildings, a gate unit is introduced to adjust the threshold value θ, which filters out weaker edge information as shown in Equation (5), resulting in edge features G. Following the same computational process used for extracting feature map E_1_, multi-scale edge features F_5_ are obtained. F_5_ is then processed in parallel through a set of three 3 × 3 convolutions, two sets of double 3 × 3 convolutions, and three sets of triple 3 × 3 convolutions to produce features $N_{i}$, where i∈(1,2,3). Subsequently, the corresponding dimensional features $M_{i}$ and $N_{i}$ are multiplied and then added together, before being concatenated along the channel dimension to produce three sets of detail features with different dimensions. Finally, three groups of 1 × 1 convolutions are applied to adjust the number of channels, thereby outputting the enhanced edge detail features $D_{i}$, where i∈(1,2,3), as shown in Equation (6).
(5)$$G=\left\{\begin{array}{c} G\left(x,y\right),G\left(x,y\right)\geq \theta \\ 0,G\left(x,y\right)<\theta \end{array}\right.$$
(6)$$D_{i}=Conv_{1\times 1}\left(Concat\left(N_{i}\cdot M_{i},N_{i}+M_{i}\right)\right)$$
where θ∈(0,1) represents the threshold value for the gate unit and *Conv* represents the convolution operation.

### 2.4. CMSCA Module

U-Net uses skip connections to reuse shallow features, but this method is a simple concatenation that ignores the semantic differences between shallow and deep features [32,33]. Consequently, this can introduce noise interference and affect the model’s segmentation performance. To further optimize the semantic differences between features of the same depth, reducing misclassification and missing classification phenomena, this research develops a new skip-connection structure named CMSCA. This structure extracts building feature information from multiple scales to enhance the network’s understanding of the data (Figure 5). By employing convolution at different scales, multi-scale feature information of buildings in shallow features is extracted and fused incrementally, preserving the semantic characteristics of targets at various scales more effectively. The CMSCA structure, inspired by the coordinated attention mechanism [34], establishes long-range dependencies between pixels. This allows the network to comprehensively integrate multi-scale feature information while enhancing the model’s ability to perceive key features of buildings.

Assuming that the input feature map F has the dimensions C × H × W (Channel × Height × Width), the feature map is processed through branch one with a 5 × 5 convolution to obtain the feature F_1_. After adding F_1_ to F, the result is passed through branch two with a 3 × 3 convolution to obtain the feature F_2_. Subsequently, F_2_ is added to F, and this output goes through branch three with a 1 × 1 convolution, resulting in the feature F_3_. The specific calculation process is expressed as follows:(7)$$F_{1}=Conv_{5\;\times 5}\left(F\right)$$
(8)$$F_{2}=Conv_{3\times 3\;}\left(F+F_{1}\right)$$
(9)$$F_{3}=Conv_{1\times 1}\left(F+F_{2}\right)$$

In branch four, the feature map F is first subjected to global average pooling separately along the width and height dimensions, generating features X and Y. After X and Y are concatenated, they go through convolution, normalization, and activation functions to produce a new feature F_4_, as shown in Equation (10).
(10)$$F_{4}=Conv[Concat\left(f_{AVG}(X),f_{AVG}(Y)\right)]$$

The feature F_4_ is split once more along the width and height to produce features X_1_ and Y_1_. These features then undergo convolution and transpose operations, followed by channel adjustment via a 1 × 1 convolution. A sigmoid function is applied to obtain weight information for both width and height. These weights are then utilized in a multiplication operation with the original feature map F to produce the feature F_5_, which emphasizes key information. This procedure is detailed in Equation (11).
(11)$$F_{5}=f_{sigmoid}[Conv(X_{1})]\times f_{sigmoid}[Conv(Y_{1})]\times F$$

Finally, by concatenating features F_1_, F_2_, F_3_, and F_4_ and then passing them through a 1 × 1 convolution to adjust the number of channels, the optimized output result Out Feature is obtained. This process is described in Equation (12).
(12)$$OUT\_Feature=Conv[Concat\left(F_{1},F_{2},F_{3},F_{4})\right)]$$
where *Conv* denotes the convolution operation; *Concat* stands for the concatenation operation; $f_{sigmoid}$ represents the sigmoid function; and $f_{AVG}$ signifies the average pooling operation.

### 2.5. Deep Supervision Mechanism

The commonly used Cross Entropy Loss Function in semantic segmentation [35] is employed to calculate the loss between the predicted outcomes and the ground truth, as shown in Equation (13).
(13)$$Loss=-\frac{1}{N}\sum_{i}\left[y_{i}log\left(p_{i}\right)+\left(1-y_{i}\right)log\left(1-p_{i}\right)\right]$$
where $y_{i}$ represents the true value of the pixel; $p_{i}$ indicates the predicted value of the pixel; and *N* denotes the total number of pixels.

To enhance the model convergence, explore the similarities and differences between pixels, and mitigate issues such as gradient vanishing and explosion in deep networks when dealing with small datasets, a deep supervision mechanism [36] is introduced. This mechanism takes different scale predictions from various stages of MFFMs (i.e., MFFM4, MFFM3, MFFM2, and MFFM1). It employs cascading addition and upsampling operations on the output feature maps to progressively restore the feature map to the original image size. Namely, the feature output from MFFM4 is upsampled and then added to the output from MFFM3, and the feature from MFFM3 is upsampled again and added to the output from MFFM2. After restoring the final output to the original image size, the loss between the ground truth and the labels is calculated through the cross-entropy loss function, and the network parameters are updated through backpropagation. The specific structure of the deep supervision module is illustrated in Figure 6.

## 3. Datasets and Evaluation Metrics

### 3.1. Dataset

#### 3.1.1. WHU Building Dataset

The WHU Building Dataset is composed of aerial and satellite datasets [37]. The aerial imagery data are sourced from the Land Information New Zealand (LINZ) website, covering approximately 450 km^2^ of Christchurch, New Zealand, with around 22,000 independent buildings. The spatial resolution is 0.3 m. Due to its wide coverage, high quality, and substantial quantity, it has become one of the most popular datasets in the remote sensing building extraction field. The aerial image dataset contains a total of 8189 images of size 512 × 512, including 4736 training images, 1036 validation images, and 2416 test images. To accelerate training and improve the generalization ability of models, data augmentation techniques are used to expand the sample size. These include horizontal flipping, vertical flipping, horizontal and vertical flipping, random erasure, brightness transformations, etc. Images are resized to 256 × 256 pixels for processing. A portion of the WHU Building Dataset is illustrated in Figure 7.

#### 3.1.2. UAV Dataset

Unmanned Aerial Vehicles (UAVs) can swiftly and accurately capture high-resolution remote sensing imagery of small areas. The UAV remote sensing data used in the study were taken by a DJI Matrice 300 drone in 2022, during various seasons. Rural architecture primarily comprises residences, granaries, and livestock sheds, which vary in shape and material composition against a diverse background. The residences are mostly one- to two-story brick and concrete buildings, featuring roofs covered in tiles or steel sheets. Granaries and livestock sheds often utilize simple steel frame structures with external coverings of steel or canvas and are irregular in shape. The imagery exhibits phenomena of the same spectrum for different substances and different spectra for the same substance, which is adverse for traditional remote sensing classification algorithms to extract buildings. The experiment selected 15 scenes of 24-bit depth Tagged Image File (TIF) format images with a spatial resolution of 0.5 m from five provinces spanning north to south China—Jilin, Hebei, Hubei, Jiangsu, and Guangxi—encompassing a total land area of 22,036,480 m^2^.

To accelerate training and enhance the model’s generalization ability, data augmentation techniques were employed to expand the sample set. These techniques include horizontal flipping, vertical flipping, both horizontal and vertical flipping, random erasure, and brightness transformation. Each image was segmented into 256 × 256 pixel tiles, resulting in a total of 6140 images (Figure 8). Finally, the images were annotated using Labelme (3.16.7) software to generate 8-bit depth mask images, and the samples were divided in a 3:1:1 ratio into a training set of 3684 images, a validation set of 1228 images, and a testing set of 1228 images.

Compared to the WHU Building Dataset, the experimental self-constructed dataset features a selection of rural buildings with diverse shapes, sizes, and structures. The style and background are more in line with the architectural characteristics of China’s rural areas. It is common to observe trees obscuring low-rise buildings, and numerous rooftops are constructed using materials such as cement slabs and tiles, which share spectral features similar to rural cement roads. The variety of architectural styles and backgrounds necessitates that the building extraction algorithms possess greater specificity and generalizability to effectively identify buildings of various sizes and to eliminate background noise. Therefore, the remote sensing images selected for this study are close to reality, better demonstrate the effectiveness of algorithms, and facilitate the application of building extraction techniques in complex rural scenarios.

### 3.2. Evaluation Metrics

The performance of the network in extracting buildings is assessed using four common semantic segmentation metrics: Precision, Recall, F1-score, and Intersection over Union (IoU). Precision is the ratio of the number of correctly predicted positive samples to the total number of samples predicted as positive. It measures the model’s accuracy in classifying positive samples, i.e., the network’s precision in recognizing buildings. Recall is the ratio of the number of correctly predicted positive samples to the total number of actual positive samples, which measures the network’s ability to correctly identify buildings. The F1-score is an indicator that takes into account both precision and recall to evaluate the network’s overall effectiveness in building recognition tasks. IoU is the ratio of the intersection to the union of the model’s predicted segmentation results and the actual segmentation results, representing the degree of overlap between the predicted segmentation and the actual segmentation. The specific formulas are as follows:(14)$$Precision=\frac{TP}{TP+FP}$$
(15)$$Recall=\frac{TP}{TP+FN}$$
(16)$$F1\text{-}score=\frac{2\times precision\times recall}{precision+recall}$$
(17)$$IoU=\frac{TP}{TP+FP+FN}$$
where *TP* (True Positive) represents the number of building pixels correctly extracted by the network; *FP* (False Positive) represents the number of background pixels misclassified as building pixels; and *FN* (False Negative) represents the number of building pixels misclassified as background pixels.

## 4. Discussion

### 4.1. Experimental Environment and Parameter Settings

The experimental environment was set up on a 64-bit Windows 11 operating system, with a CPU of 13th Gen Intel(R) Core(TM) i7-13700KF and a GPU of GeForce RTX 4070 (with 12 GB of video memory). The programming environment was Python 3.10, mainly utilizing open-source Python libraries such as PyTorch 11.8, NumPy, and Matplotlib.

The hyperparameter settings for the experiments are outlined as follows: The GPU’s batch size was set to 4, with a decay strategy of cosine annealing, and the Adam (Adaptive Moment Estimation) optimizer was used. The initial learning rate was set to 0.0001, and the model was trained for a total of 200 epochs. Additionally, to improve training efficiency and prevent overfitting, an early stopping strategy was introduced. This involved monitoring the model’s performance on the validation set and halting training if the performance evaluation index did not improve for more than 15 rounds. Lastly, we utilized color space transformation as a data augmentation technique to improve the model’s generalization ability and robustness.

### 4.2. Experiment Results

#### 4.2.1. Building Extraction Results for UAV Dataset

Figure 9 shows the building extraction results of MAD-UNet on the UAV dataset. In this study, typical images from rural areas in different provinces were selected for testing to verify the effectiveness of MAD-UNet in extracting buildings from various rural regions. From the (a), (b), and (c) scenarios, MAD-UNet demonstrates excellent recognition performance for medium to large buildings, being able to extract most buildings completely and effectively segment the gaps between buildings, while avoiding noise interference from trees, shadows, and other elements between buildings in the background. When dealing with small buildings, especially in dense and cluttered scenes, as shown in the (e) and (d) scenarios, MAD-UNet can accurately retrieve the spatial location of buildings, preventing most cases of misclassification and missed detection.

Table 2 presents the quantitative results of building extraction in rural areas by MAD-UNet across different regions. For the evaluation, thirty images were randomly selected from each of the five provinces—Jilin, Hebei, Hubei, Jiangsu, and Guangxi—and were divided into five groups representing rural scenes from different geographical locations. The average Precision, Recall, F1-Score, and IoU were calculated for images in each region to carry out the quantitative assessment. The results indicated that the Precision metric for all five groups tested exceeded 85%, demonstrating that MAD-UNet is capable of accurately identifying buildings against the complex rural background. Notably, in groups 1, 3, and 4, where the building size varied and the layout was relatively orderly, IoU surpassed 77%. This suggests that MAD-UNet performs well in recognizing buildings of various scales. In groups 3 and 5, the IoU metric decreased by 5%, likely because these groups had denser building distributions with more chaotic layouts and increased noise factors. Furthermore, the F1-score for all test data maintained above 84%, further validating the robustness and accuracy of this method in dealing with diverse rural scenes.

#### 4.2.2. Building Extraction Results for WHU Building Dataset

Figure 10 displays the extraction results of MAD-UNet on the WHU Building Dataset. Six images were randomly selected for testing to comprehensively evaluate the model’s performance. The third column in the figure shows the predictive results of MAD-UNet, where rows (a)–(c) illustrate that the model effectively segments medium to large buildings, overcoming the false segmentation phenomenon of internal voids in large buildings caused by the shadows on their rooftops. In row (c), which contains images without buildings, MAD-UNet correctly delineates the roads and shadows that share similar spectral information with buildings. Rows (d)–(e) showcase the extraction results for small-sized buildings. MAD-UNet achieved satisfactory segmentation in the presence of noise factors such as trees, shadows, and roads between buildings. In brief, these results prove that the integration of multiple scale features in MAD-UNet significantly enhances its ability to model buildings of various sizes and accurately extract building information amidst various noise factors.

Table 3 quantitatively evaluates the extraction results of MAD-UNet on buildings of varying sizes. According to different building scales, 30 images of large and medium-sized buildings with open scenes, and 30 images of small to medium-sized buildings with many tree shadows were randomly selected from the testing set. In the task of extracting large and medium-sized buildings, MAD-UNet achieved 96.74% Precision, 96.44% F1-score, and 93.13% IoU; in the task of small to medium-sized building extraction, it achieved 95.08% Precision, 94.59% F1-score, and 89.74% IoU. The precision of MAD-UNet exceeded 95%, indicating that the model can accurately differentiate between buildings and non-building backgrounds in urban areas. Moreover, for both groups of test data, the precision of buildings was higher than 95%, and the F1-score surpassed 94%, further demonstrating the model’s high robustness in the face of different scales and densely distributed building scenes, showcasing its adaptability to complex scenarios.

### 4.3. Comparative Experimental Results

To further validate the capability of MAD-UNet in extracting rural buildings, a comparative analysis was conducted against five semantic segmentation networks, including U-Net [9], PSPNet [10], DeepLabV3+ [15], HRNet [38], ISANet [39], and AGSCNet [18]. Brief introductions to these models are illustrated as follows:

(1) Proposed by Ronneberger et al. [9], U-Net utilizes a skip-connection structure that links feature maps from the encoder with corresponding feature maps in the decoder. This allows the network to utilize a richer set of low-level features. It requires less data, trains quickly, and has been widely applied across various domains for semantic segmentation tasks, which is why it has been selected as a benchmark model;

(2) PSPNet is a semantic segmentation network introduced by Zhao et al. [10]. Its core concept is to incorporate a pyramid pooling module, which allows the network to perceive image information at multiple scales, thereby enhancing its ability to understand large-scale scenes;

(3) DeepLabV3+ [15] employs atrous convolution to obtain a larger receptive field without increasing the number of parameters and computational cost. The decoder further merges low-level and high-level features, effectively handling multi-scale information while maintaining high resolution. This approach has achieved significant success in many image segmentation tasks;

(4) Proposed by Sun et al. [38], HRNet maintains multi-scale information across different branches simultaneously, enabling a more comprehensive understanding of details and global structures in an image. It is widely used in tasks such as human pose estimation, keypoint detection, and semantic segmentation;

(5) Introduced by Huang et al. [39], ISANet is aimed at reducing the computational burden of attention mechanisms. It illustrates an Interlaced Sparse Self-Attention mechanism that establishes long-range dependencies between pixels in an image to expand the receptive field. Its effectiveness has been confirmed on multiple semantic segmentation datasets;

(6) AGSCNet [18] builds upon the U-Net architecture by integrating multiple attention gate modules and a context collaboration network structure. It has achieved good results in tasks involving the extraction of buildings in various rural areas of China.

#### 4.3.1. Comparison Experiment on UAV Dataset

Figure 11 displays the extraction results from seven different semantic segmentation models on the private UAV dataset. The analyzed regions are marked by rectangular boxes in the images, where red boxes signify misclassification issues, yellow boxes point to omission problems, and blue boxes emphasize unclear demarcations of building boundaries.

Located in northeastern China, Jilin Province endures harsh winters with significant snowfall, requiring specific architectural adaptations for rural buildings. To prevent snow accumulation, these structures often have sloping roofs made of corrugated steel, a material chosen for its durability and ease of maintenance. However, this design leads to a dense array of similar roofs that, along with the shadows they cast, complicates the delineation of individual building boundaries. In Figure 11, the blue rectangular box highlights the issue of unclear building boundaries, with PSPNet demonstrating the poorest performance in discriminating shadows between buildings. U-Net, DeepLabV3+, HRNet, ISANet, and AGSCNet show commendable capabilities in distinguishing shadows in larger gaps between buildings. However, their performance declines with the identification of shadows in densely spaced buildings.

In northern China’s Hebei Province, rural buildings predominantly feature roofs made of brick and tile, which present a gray-white hue that resembles the spectra of roads and courtyard surfaces. This similarity often leads to misclassification errors when distinguishing these structures from their surroundings. In Figure 11, the red rectangular box indicates misclassification issues, with PSPNet, DeepLabV3+, HRNet, ISANet, and AGSCNet often mistaking the roads for buildings. While U-Net is proficient at differentiating roads from buildings, it may struggle to classify courtyard surfaces accurately within a quadrangle-style layout, potentially leading to classification errors.

Central China’s Hubei Province is home to rural buildings with roofs primarily made from brick and tile. These buildings are typically arranged in a courtyard-style layout, sparsely distributed across the landscape, and exhibit significant variation in size. The spectral similarity between the roofing materials and the surfaces of roads or bare ground often results in misclassification issues. In Figure 11, the red rectangular box depicts these misclassification issues, with U-Net, DeepLabV3+, HRNet, ISANet, and AGSCNet incorrectly identifying roads. Additionally, the considerable differences in building sizes can lead to omissions, as shown in Figure 11, where the yellow rectangular box indicates that PSPNet, U-Net, DeepLabV3+, HRNet, ISANet, and AGSCNet all show varying degrees of omission, especially for smaller buildings.

In the eastern coastal Jiangsu Province and the southwestern Guangxi Province, both of which experience a rainy climate, rural buildings are designed with sloping roofs to facilitate water runoff. These buildings tend to be smaller and densely clustered, which often results in omission issues. For scenes in Jiangsu and Guangxi (Figure 11), the yellow rectangular box points to omission problems where PSPNet, U-Net, DeepLabV3+, HRNet, ISANet, and AGSCNet all exhibit varying degrees of omission for small and densely grouped rural buildings.

Table 4 presents the accuracy results of building extraction for the seven networks. MAD-UNet achieved the highest precision across all four evaluation metrics: Precision, Recall rate, F1-score, and IoU, with scores of 88.39%, 86.19%, 87.28%, and 77.43%, respectively. When compared to the baseline network U-Net, these scores represent improvements of 0.2%, 2.95%, 2.12%, and 2.4%, respectively. This demonstrates that the improved methodology proposed in this study effectively enhances model accuracy.

Synthesizing the aforementioned quantitative and qualitative results, in the task of extracting rural buildings from different areas, MAD-UNet effectively mitigates the influences of different land covers and shadows, ameliorating misclassification and omissions, and achieving superior boundary integrity compared to other networks. PSPNet and DeepLabV3+, using feature pyramid-based approaches aimed at processing features of varying scales, lacked precision in local information, leading to information loss when addressing small targets. Therefore, they demonstrated the poorest performance in extracting small-scale objectives. U-Net and HRNet had fewer omissions and misclassifications in their extraction results but faced issues with edge adhesion, which blurred boundaries between adjacent buildings. It is hypothesized that the extensive use of convolutions and pooling operations in U-Net and HRNet hinders the networks’ ability to accurately restore the original spatial structure, leading to blurred and adhesive building boundaries. ISANet, designed to reduce the computational burden of attention mechanisms while establishing long-range relationships between pixels, excessively focuses on edge connectivity at the expense of clear delineation between buildings, resulting in boundary adhesion. AGSCNet’s CC-Net adopts atrous rates of (1, 6, 12, 18) within ASPP, which for rural small-sized buildings may be overly large, prompting the model to focus too much on contextual information and thus neglecting the details of small-scale targets, causing omissions.

#### 4.3.2. Comparison Experiment on WHU Building Dataset

Figure 12 displays the extraction results of the seven semantic segmentation models on the WHU Building Dataset. Rows 1 to 6 are arranged in ascending order of building sizes, and areas of interest are identified with rectangular frames—red frames for misclassification issues, yellow for omissions, and blue for unclear boundary distinctions. PSPNet, DeepLabV3+, HRNet, ISANet, AGSCNet, and U-Net all experience a certain degree of misclassification, omission, and unclear boundary demarcation when extracting buildings. In the first row, buildings within the red frame share similar spectral features with the roads in the background, leading to misclassification by all algorithms except MAD-UNet. In the second row, which presents buildings of varying sizes, DeepLabV3+ and HRNet completely overlook the small building within the yellow frame, while MAD-UNet exhibits a more complete segmentation of this small building compared to PSPNet, ISANet, AGSCNet, and U-Net. In the third row, the small building inside the yellow frame is affected by the surrounding tree shadows, causing a certain degree of omission by all models except MAD-UNet. For the fourth and fifth rows, MAD-UNet surpasses other models in smoothly delineating the large building contours within the blue frames, achieving clearer boundary demarcation between buildings.

Table 5 displays the quantitative comparison results of 10 different semantic segmentation models, among which B-FGC-Net [19], AGs-UNet [17], and Chen’s model [40] are the latest algorithmic models in the field of building extraction in recent years. MAD-UNet achieved the optimal values in three evaluation metrics: Precision, F1-score, and IoU. The performance enhancements of MAD-UNet relative to other models, including PSPNet, HRNet, DeepLabV3+, ISANet, AGSCNet, U-Net, B-FGC-Net, Chen’s model, and AGs-UNet, are detailed in Table 6.

A Precision of 95.98% indicates that there are fewer mistakes among pixels labeled as buildings, an IoU of 91.02% shows that the segmentation structures match the actual real building locations, and an F1-score of 95.31% reflects a well-balanced trade-off between Precision and Recall rate, demonstrating high overall performance. It further validates quantitatively the robustness of MAD-UNet, confirming its capability to handle building extraction tasks in a variety of complex scenarios.

### 4.4. Melting Experimental Results

To evaluate the effectiveness of each component in MAD-UNet, ablation studies were conducted with different combinations of modules, including DEM, MFFM, ASPP, and CMSCA. Because MFFMs and DEMs need to be utilized together, they were grouped into one experimental group, yielding a total of four groups for the ablation experiments. ResNet50 served as the backbone feature extraction network for each group.

The experimental results are presented in Table 7. In Experiment 2, which incorporated only the ASPP module, there were improvements in the Recall rate and F1-score, but the increase in false positives resulted in a decrease in Precision and IoU. This suggests that employing only the ASPP module has its limitations. Although ASPP can broaden the receptive field by integrating multi-scale context information, the influx of excessive background information caused model misjudgments. Experiment 3, which added the MFFM module to the setup of Experiment 2, demonstrated increases of 0.97%, 1.53%, and 1.27% in Precision, Recall rate, and F1-score, respectively. This indicates that the addition of MFFM significantly boosts the model’s ability to identify positive samples. An IoU increase of 1.95% also suggests that the model’s predictions are more aligned with the actual targets. By merging multi-scale features and enhancing details, MFFM effectively captures the complex structures and intricate features of buildings, thus improving the model’s precision in recognizing and locating positive samples. In Experiment 4, which integrated the CMSCA module into Experiment 3, there was a slight decrease in Precision. However, there were increases of 1.39%, 0.57%, and 0.89% in Recall rate, F1-score, and IoU, respectively. This indicates that incorporating the CMSCA module into the skip connections of MAD-UNet enables the model to effectively capture both global and local information, bridging semantic gaps with deep feature maps. The coordinated attention mechanism introduced helps to suppress ineffective noise and enhances the model’s resilience to interference. This makes the model more adaptable to complex environments and more precise in focusing on target areas, thus improving its ability to identify positive samples. In Experiment 5, based on Experiment 4, the ASPP module was removed. This led to decreases in Recall, F1-score, and IoU, particularly in the Recall rate, which suggests a weakened ability of the model to identify positive samples. Compared with Experiment 4, the overall decrease in Recall rate, F1-score, and IoU, along with the reduced Precision, further supports the notion that while the ASPP module bolsters the model’s capacity to recognize multi-scale features, it also introduces some noise from the background. This leads to certain misjudgments and results in a slight decrease in precision.

## 5. Conclusions

To tackle the challenges of misclassification, omissions, and suboptimal edge extraction in rural residential structures identified within high-resolution UAV remote sensing images, an enhanced network model built upon the U-Net framework, named MAD-UNet, was introduced. This model integrates ResNet50 and atrous convolutional pyramid pooling in the decoder to enhance the extraction of global and multi-scale features. It also utilizes a cascade multi-scale convolution module and a coordinated attention mechanism in the skip connections to bridge semantic gaps at equivalent feature map depths and to mitigate irrelevant background noise. The decoder combines MFFMs and DEMs to ensure a comprehensive representation of the semantic information of buildings while preserving detail features. The model was tested on 0.5 m resolution UAV aerial images taken across different seasons in 2022 from various provinces in China, including Jilin, Hebei, Hubei, Jiangsu, and Guangxi, and on the publicly accessible WHU Building Dataset. Comparative analyses were performed with models such as U-Net, PSPNet, HRNet, DeepLabV3+, ISANet, and AGSCNet. MAD-UNet demonstrated enhancements in reducing misclassification and omissions, as well as in boundary extraction, with Precision, Recall rate, F1-score, and IoU achieving 88.39%, 86.19%, 87.28%, and 77.43%, respectively. Ablation experiments further verified the contribution of the ASPP, CMSCA, and MFFM modules to the overall efficacy of the MAD-UNet model. The research findings attest that the MAD-UNet model can effectively address issues of missed detection, false detection, and boundary adhesion in the extraction of rural buildings across diverse regions of China. Moreover, the model proves to be apt for the extraction of rural buildings in these varied geographical areas. However, the MAD-UNet model is computationally intensive due to its intricate structure. Future efforts will be directed towards simplifying the model through methods such as distillation and pruning [41] to diminish its complexity, reduce the number of parameters, and shorten training duration. Additionally, semi-supervised learning [42] will be explored to fortify the model’s generalization ability by incorporating more unlabeled data, thereby improving its adaptability to complex settings. This strategy aims to facilitate the efficient deployment of the MAD-UNet model in practical applications with constrained computational resources and to enhance its proficiency in processing novel scenes. The goal is to deliver a reliable technological approach for rural building extraction tasks that effectively balances performance and efficiency.

## Figures and Tables

**Figure 1 sensors-24-02393-f001:**
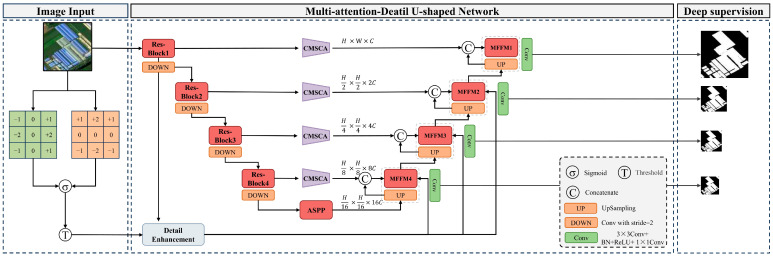
MAD-UNet structural diagram.

**Figure 2 sensors-24-02393-f002:**
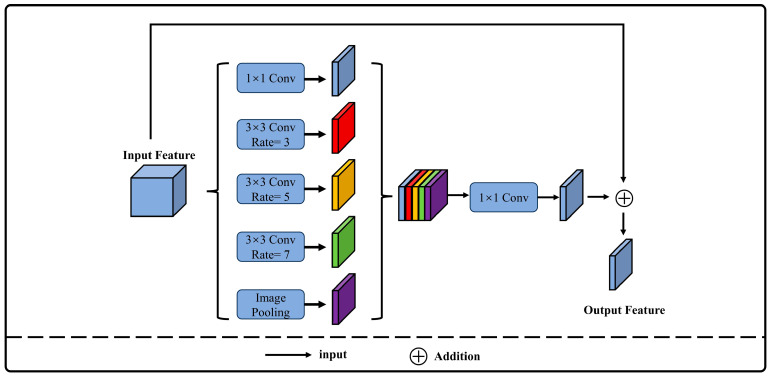
Architecture of atrous convolutional pyramid pooling module.

**Figure 3 sensors-24-02393-f003:**
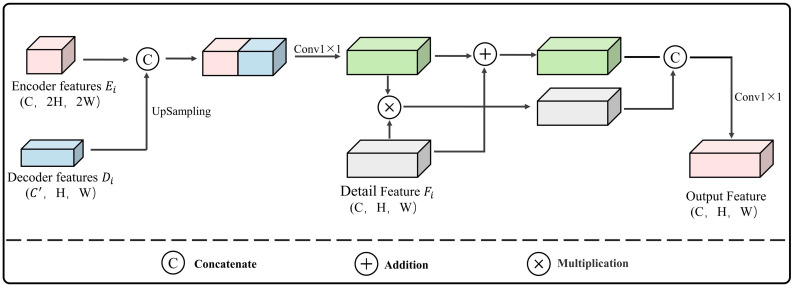
Architecture of MFFM.

**Figure 4 sensors-24-02393-f004:**
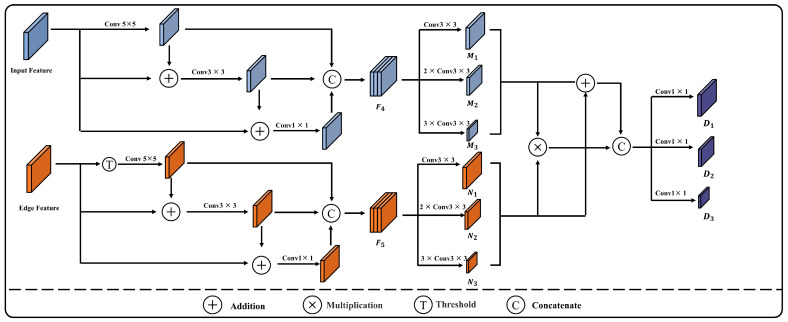
Architecture of DFEM.

**Figure 5 sensors-24-02393-f005:**
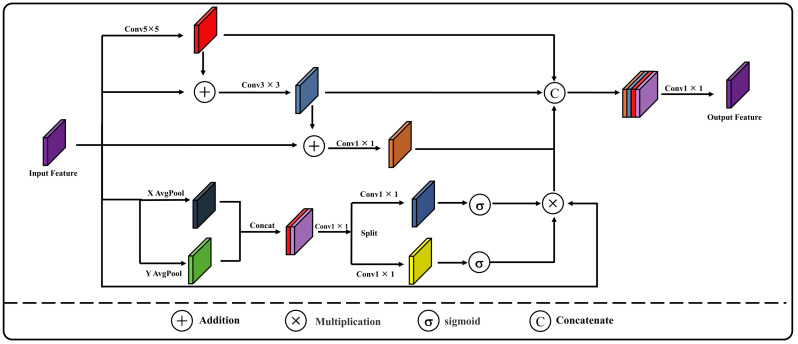
Architecture of CMSCA.

**Figure 6 sensors-24-02393-f006:**
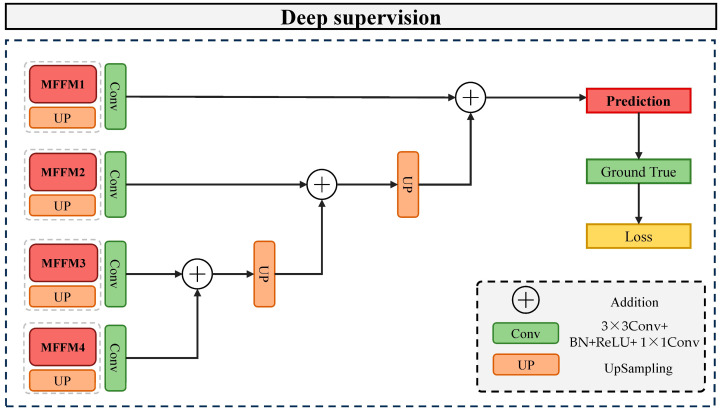
Architecture of extraction module.

**Figure 7 sensors-24-02393-f007:**
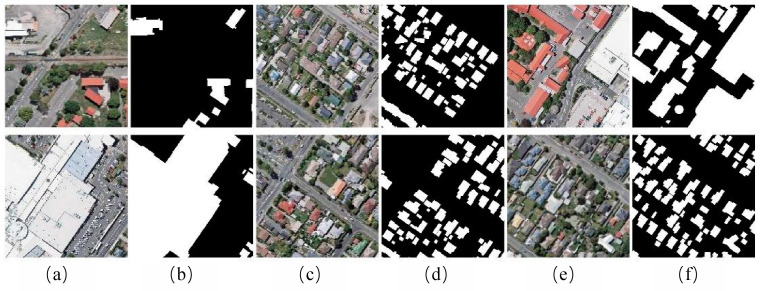
Paired images of WHU Building Dataset: (**a**,**c**,**e**) Actual image; (**b**,**d**,**f**) Corresponding labels.

**Figure 8 sensors-24-02393-f008:**
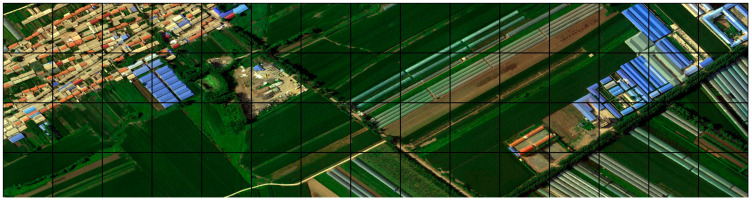
Cropped and tiled sections of imagery in 256 × 256 pixels highlighted by black borders.

**Figure 9 sensors-24-02393-f009:**
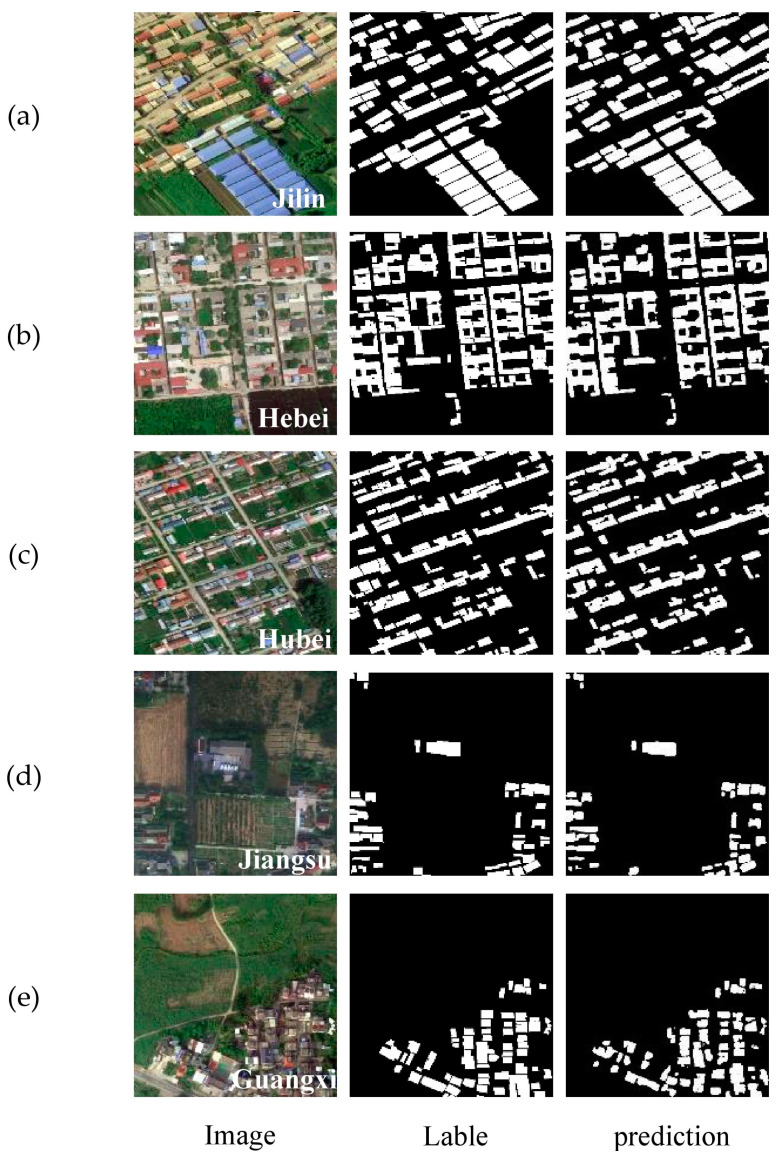
Extraction results on the UAV dataset:(**a**) image is of rural buildings in Jilin Province. (**b**) image is of rural buildings in Hebei Province. (**c**) image is of rural buildings in Hubei Province. (**d**) image is of rural buildings in Jiangsu Province. (**e**) image is of rural buildings in Guangxi Province.

**Figure 10 sensors-24-02393-f010:**
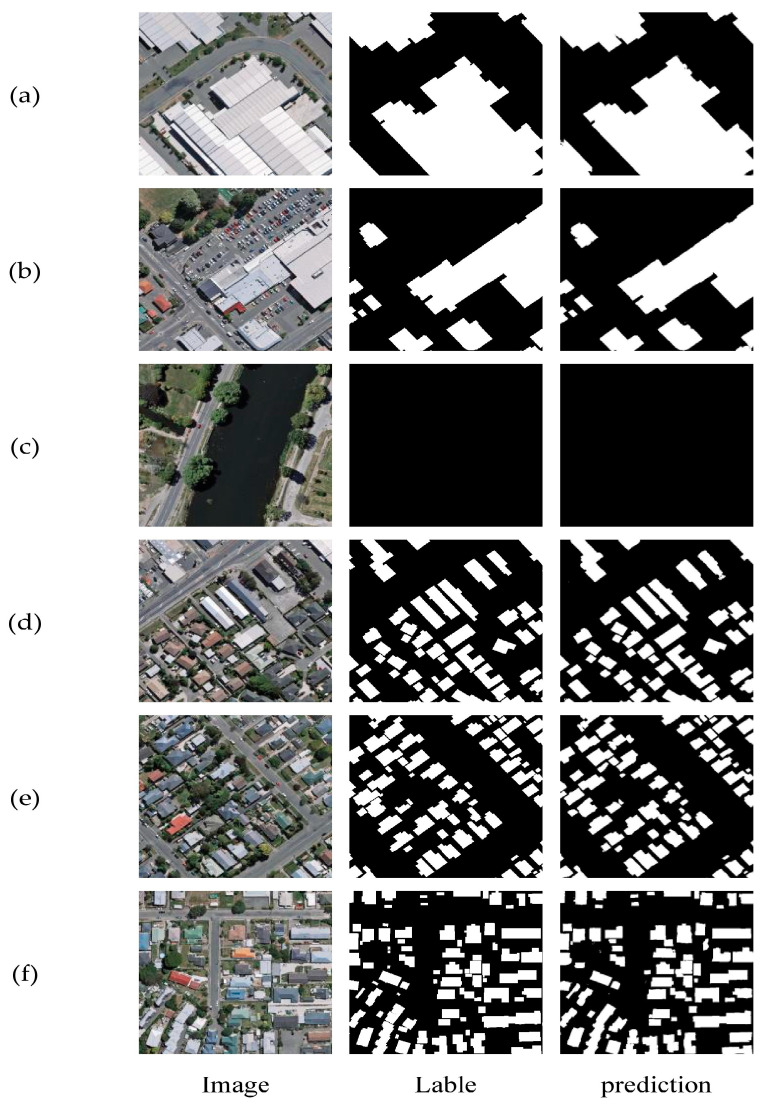
Extraction results on the WHU Building Dataset: (**a**,**b**) Images of large and medium-sized buildings; (**c**) Image with trees, shadows, and roads; (**d**–**f**) Images of small buildings.

**Figure 11 sensors-24-02393-f011:**
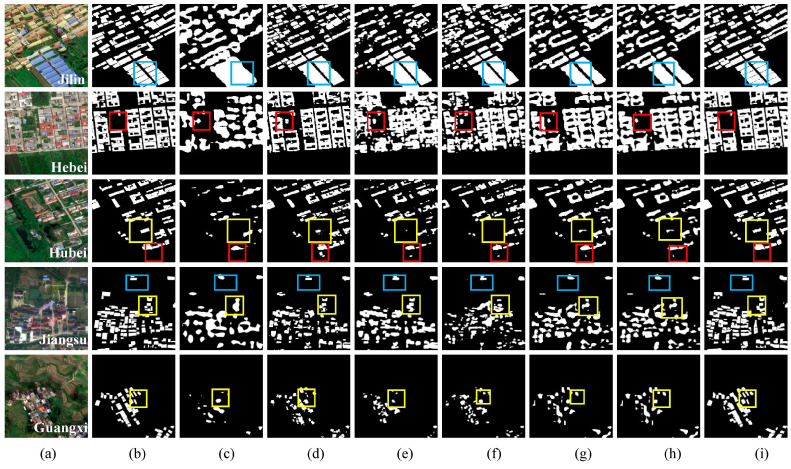
Comparative results on the UAV Dataset; (**a**) Original image; (**b**) Ground truth label; (**c**) PSPNet; (**d**) DeepLabV3+; (**e**) HRNet; (**f**) ISANet; (**g**) AGSCNet; (**h**) U-Net; (**i**) MAD-UNet.

**Figure 12 sensors-24-02393-f012:**
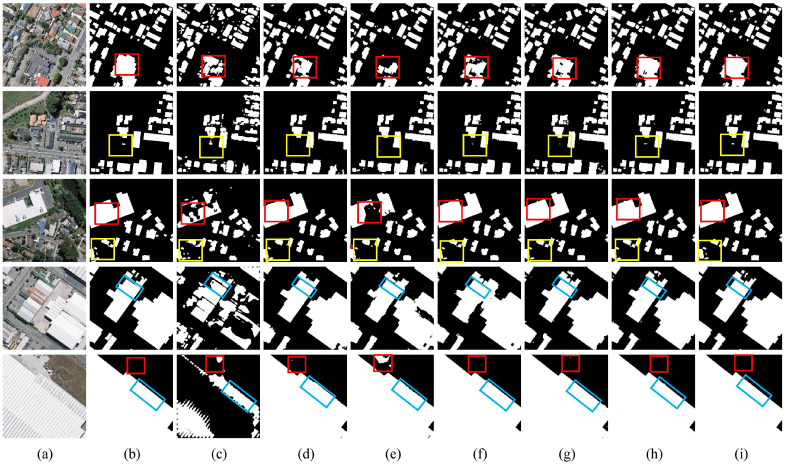
Comparative results on the WHU Building Dataset: (**a**) Original image; (**b**) Ground truth label; (**c**) PSPNet; (**d**) DeepLabV3+; (**e**) HRNet; (**f**) ISANet; (**g**) AGSCNet; (**h**) U-Net; (**i**) MAD-UNet.

**Table 1 sensors-24-02393-t001:** Comprehensive parameter of MAD-UNet model.

Stage	Template	Output Size
Input		3 × 256 × 256
1	1 × 1Conv + BN + ReLU	64 × 256 × 256
	3 × Res-Block	
	CMSCA	
2	4 × Res-Block	128 × 128 × 128
	CMSCA	
3	6 × Res-Block	256 × 64 × 64
	CMSCA	
4	3 × Res-Block	512 × 32 × 32
	CMSCA	
Bridge	ASPP	1024 × 16 × 16
5	Upsampling	512 × 32 × 32
	Concat + 1 × 1Conv + BN + ReLU	
	MFFM1	
Deep supervision	3 × 3Conv + BN + ReLU + 1 × 1Conv	2 × 32 × 32
6	Upsampling	256 × 64 × 64
	Concat + 1 × 1Conv + BN + ReLU	
	MFFM2	
Deep supervision	3 × 3Conv + BN + ReLU + 1 × 1Conv	2 × 64 × 64
7	Upsampling	128 × 128 × 128
	Concat + 1 × 1Conv + BN + ReLU	
	MFFM3	
Deep supervision	3 × 3Conv + BN + ReLU	2 × 128 × 128
	1 × 1Conv	
8	Upsampling	64 × 256 × 256
	Concat + 1 × 1Conv + BN + ReLU	
	MFFM4	
Prediction	3 × 3Conv + BN + ReLU	2 × 256 × 256
	1 × 1Conv	

Note: CMSCA represents skip-connection structure; MFFM denotes Multiscale Feature Fusion structure; Conv indicates convolution operation; BN stands for Batch Normalization; Concat stands for the concatenation operation.

**Table 2 sensors-24-02393-t002:** Quantitative results of the UAV dataset.

Group	Precision	Recall	F1-Score	IoU
1	88.19%	91.40%	89.77%	81.44%
2	85.32%	82.97%	84.13%	72.61%
3	86.34%	89.89%	88.07%	78.69%
4	86.81%	82.93%	84.83%	73.65%
5	89.39%	85.92%	87.62%	77.97%

**Table 3 sensors-24-02393-t003:** Quantitative results of the WHU Building Dataset.

Group	Precision	Recall	F1-Score	IoU
1	96.74%	96.14%	96.44%	93.13%
2	95.08%	94.11%	94.59%	89.74%

**Table 4 sensors-24-02393-t004:** Accuracy comparative results of the UAV dataset.

Model	Precision	Recall	F1-Score	IoU
PSPNet	66.51%	45.09%	43.24%	37.51%
HRNet	77.75%	67.03%	63.23%	56.24%
DeepLabV3+	73.49%	61.90%	67.20%	50.60%
ISANet	75.75%	67.23%	71.24%	55.32%
AGSCNet	86.32%	77.16%	81.49%	68.76%
U-Net	88.19%	83.24%	85.16%	75.03%
MAD-UNet	88.39%	86.19%	87.28%	77.43%

**Table 5 sensors-24-02393-t005:** Accuracy comparative results of the WHU Building Dataset.

Model	Precision	Recall	F1-Score	IoU
PSPNet	85.48%	51.67%	58.61%	63.58%
HRNet	92.57%	88.83%	90.68%	82.31%
DeepLabV3+	93.11%	92.99%	93.05%	85.66%
ISANet	89.23%	82.47%	75.89%	85.72%
AGSCNet	94.20%	92.94%	93.55%	87.97%
U-Net	87.39%	90.06%	91.53%	87.39%
Chen’s	93.25%	95.56%	94.40%	89.39%
B-FGC-Net	95.03%	94.49%	94.76%	90.04%
AGs-UNet	93.70%	-	-	85.50%
MAD-UNet	95.98%	94.64%	95.31%	91.02%

Note: “-” indicates that the original literature did not provide the data.

**Table 6 sensors-24-02393-t006:** Quantitative Performance Gains of MAD-UNet Over Other Models.

Model	Precision	Recall	F1-Score	IoU
PSPNet	+10.5%	+42.97%	+36.7%	+27.44%
HRNet	+3.41%	+5.81%	+4.63%	+8.71%
DeepLabV3+	+2.87%	+1.65%	+2.26%	+5.36%
ISANet	+6.75%	+12.17%	+19.42%	+5.30%
AGSCNet	+1.78%	+1.70%	+1.76%	+3.05%
U-Net	+8.95%	+4.58%	+3.78%	+3.63%
Chen’s	+2.73%	−0.92%	+0.91%	+1.63%
B-FGC-Net	+0.95%	+0.15%	+0.55%	+0.98%
AGs-UNet	+2.28%	-	-	+5.52%

Note: “-” indicates that the original literature did not provide the data.

**Table 7 sensors-24-02393-t007:** Accuracy comparative results of melting experiment.

No.	Module			Evaluation Metric			
	ASPP	CMSCA	MFFM	Precision	Recall	F1-Score	IoU
1				88.19%	83.24%	85.16%	75.03%
2	√			87.73%	83.27%	85.44%	74.59%
3	√		√	88.70%	84.80%	86.71%	76.54%
4	√	√	√	88.39%	86.19%	87.28%	77.43%
5		√	√	88.75%	84.65%	86.65%	76.45%

## Data Availability

The WHU Building Dataset is publicly accessible at http://gpcv.whu.edu.cn/data/building_dataset.html, accessed on 22 January 2023. For the private UAV datasets, requests for access can be directed to xuehang@stumail.nciae.edu.cn.

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
