# Peer review of "MAD-UNet: A Multi-Region UAV Remote Sensing Network for Rural Building Extraction"

_sensors, 2024, doi:10.3390/s24082393_

Round 1

Reviewer 1 Report

Comments and Suggestions for Authors

Minor comments

Check the first paragraph of the introduction, lines 37 and 42… “resulting in leading to suboptimal.”

Comments on the Quality of English Language

Quality of English Language is appropraite

Reviewer 2 Report

Comments and Suggestions for Authors

1. In the ablation experiment, the addition of ASPP module alone led to a decrease in IoU. The use of ASPP, CMSCA, and MFFM modules effectively improved the results of building extraction, but it cannot be proven that the simultaneous use of ASPP and other modules had a beneficial effect on the results. Can more ablation experiments be added (such as the experimental results of other modules without ASPP) to prove the effectiveness of using ASPP.

2. In the experimental setup, please explain whether this article used data augmentation and other methods during the model training process.

3. This article compares the visualization results in experiments to demonstrate the excellent performance of the proposed method. Can more visualization experiments be added (such as feature maps obtained from images after passing through the module proposed in this article, in order to more intuitively see the feature extraction effect of the module).

4.The theme of this article is the construction of beautiful rural areas and the extraction of rural buildings. However, in the example image in Figure 7, we should see a town with dense buildings. Please explain why this dataset is used.

5. Please revise the layout of the article, such as Table 5

Comments on the Quality of English Language

The English expression in this article needs further refinement and optimization. It is recommended that native speakers revise it

Reviewer 3 Report

Comments and Suggestions for Authors

Comments on the Quality of English Language

Reviewer 4 Report

Comments and Suggestions for Authors

The authors have done a good job presenting useful research results.

Contribution: A multi-scale fusion and detail enhancement network for rural building extraction. The results demonstrate its efficacy for extracting rural buildings from UAV remote sensing images across different areas.

I suggest you address "major issues" and revise minor if you are revising to resubmit.

Major issues:

You have done a good job of presenting the comparative results in Figures 11 and 12. That said, for someone less familiar with this research area, it is still somewhat challenging to spot all the differences. I have no recommendation, but consider if there is additional graphical illustration or text discussion to help out. It is a key aspect of the paper.  Finally, what you have may be fine.

Lines 546 - 550 may be more effectively presented in table form.

Minor issues:

l36: "remote sensing images [2]," -> "remote sensing images [2]."

l42: "resulting in leading to" -> "resulting in"

l90: "atrous" -> "atrous (dilated)"

l112: "proprietary UAV", spell out all acronyms and abbreviations on first use

l139: "diagra" -> "diagram"

l151: "optimiz" -> "optimize"

Table 1, last row: Is "Prection" correct? "Precision"? "Prediction"?

l179: "Figure 2. Architecture of dilated convolution pyramid pooling module", generally you have used "atrous", so probably use "atrous" here instead of "dilated"

Be consistent (and comply with journal guidelines) for figure refs. Sometimes you say "Figure" (e.g. line 173), sometimes you say "Fig" (e.g. line 188)

Figure 3, "OutPut" -> "Output". Also, be consistent in word capitalization. True for all figures.

l203: "which is subsequently added to which is subsequently added to" does not make sense, needs rewording

Suggest you only need to define things like conv and cat (and similar) once in paper

l273: "To enhance the model converge" -> "To enhance the model convergence" (I think)

Don't mix your notation for number format. Line 318 you have 22.03648, while line 326 and elsewhere you have 1,228. Probably use commas throughout.

l339: "as follows:" -> "as follows."

l398: "results of" -> "results for"

For table 2, are all values percent? If so, include % for each

l414 "results of" -> "results for"

l514: "of seven" -> "of the seven"

Reviewer 5 Report

Comments and Suggestions for Authors

The study devised a novel MAD-UNet model for rural buildings, adept at extracting buildings of various scales and delineating their edge information effectively. But there are still some suggestions:

(1) In the realm of remote sensing, DEM typically refers to digital elevation model. To prevent confusion, it is advisable to employ DFEM as the abbreviation for detail feature enhancement module.

(2) When it comes to edge extraction, Canny typically outperforms Sobel. Why then was Sobel chosen for the DEM?

(3) In terms of the paper structure, it is suggested to divide the discussion into two parts: Experiments and Discussion. Sections 4.1, 4.2, and 4.3 will serve as Experiments, while Section 4.4 will be the Discussion section. Furthermore, the Discussion section should address the shortcomings of the model and the impact of key parameter selections on the model, such as the effect of changes in θ in Sobel on the model results. Additionally, considering the author's mention of drone images being available in various seasons, perhaps discussing the model's robustness across different seasons would be beneficial, this is just a suggestion.

(4) In Section 4.2.1, qualitatively, it's not easy for readers to discern the sizes of buildings in images across different scenes. It is suggested to mention the approximate sizes of buildings in the first three scenes and the last two scenes in the paper. The buildings in the latter two scenes appear to be less dense, indicating they might not be located in areas with densely populated structures typical of villages. Quantitatively, the author randomly selected 30 images for calculating quantitative indicators. If some of these images do not feature buildings, it could potentially impact the evaluation results. It is recommended to use a larger set of images, preferably including all predicted results containing buildings.

(5) In Section 4.4, it would also be valuable to present the individual contributions of the CMSCA and MFFM modules to the model's performance.

Comments on the Quality of English Language

English language slightly modified.

Round 2

Reviewer 3 Report

Comments and Suggestions for Authors

Please see the attached document

Reviewer 5 Report

Comments and Suggestions for Authors

The author responded to all my comments and I agree to accept this manuscript in its current version.

Author Response

Thank you sincerely for your affirmation of our paper. It's incredibly gratifying to see our efforts recognized and accepted.